# Revisiting the Activation Function for Federated Image Classification

**Jaewoo Shin, Taehyeon Kim, Se-Young Yun**
KAIST AI
Seoul, Korea
{yimsungen5, potter32, yunseyoung}@kaist.ac.kr

## Abstract

Federated learning (FL) has become one of the most popular distributed machine learning paradigms; these paradigms enable training on a large corpus of decentralized data that resides on devices. The recent evolution in FL research is mainly credited to the refinements in training procedures by developing the optimization methods. However, there has been little verification of other technical improvements, especially improvements to the activation functions (e.g., ReLU), that are widely used in the conventional centralized approach (i.e., standard data-centric optimization). In this work, we verify the effectiveness of activation functions in various federated settings. We empirically observe that off-the-shelf activation functions that are used in centralized settings exhibit a totally different performance trend than do federated settings. The experimental results demonstrate that HardTanh achieves the best accuracy when severe data heterogeneity or low participation rate is present. We provide a thorough analysis to investigate why the representation powers of activation functions are changed in a federated setting by measuring the similarities in terms of weight parameters and representations. Lastly, we deliver guidelines for selecting activation functions in both a *cross-silo* setting (i.e., a number of clients $\leq 20$) and a *cross-device* setting (i.e., a number of clients $\geq 100$). We believe that our work provides benchmark data and intriguing insights for designing models FL models. The code is available at https://github.com/Jaewoo-Shin/FL_ACT.

## 1 Introduction

Federated learning (FL) has become a common and ubiquitous paradigm for collaborative machine learning techniques [3, 4, 23, 33, 34, 44, 37, 45] because it maintains data privacy. Each client (e.g., mobile devices or the whole business) communicates with the central server by transferring the model but not the data; all local updates are aggregated in a global server-side model. Although a centralized method enhances generalization by employing a large amount of training data, the features of the FL methods appear to differ from those of a centralized method owing to data heterogeneity, client resource capability, and model communication [24, 37, 50].

Most FL studies focus on improving the performance of the global model. To this end, they apply a new regularizer in the optimization algorithm [1, 25, 32, 33, 18, 35, 46, 47, 48, 50, 21, 10, 13]. Recently, there has been an increasing demand for the personalization of models according to the client. Jiang et al. [22] and Fallah et al. [9] attempt to train personalized models for each client with a few rounds of fine-tuning rather than focusing on the performance of a server model. In consideration of system heterogeneity (i.e., clients having different computational and communication capabilities), Avdiukhin et al. [2] mitigate model communication problems by using asynchronous local SGD; Horvath et al. [17] improve accuracy in heterogeneous resource capacity by using different model sizes per client.

36th Conference on Neural Information Processing Systems (NeurIPS 2022).

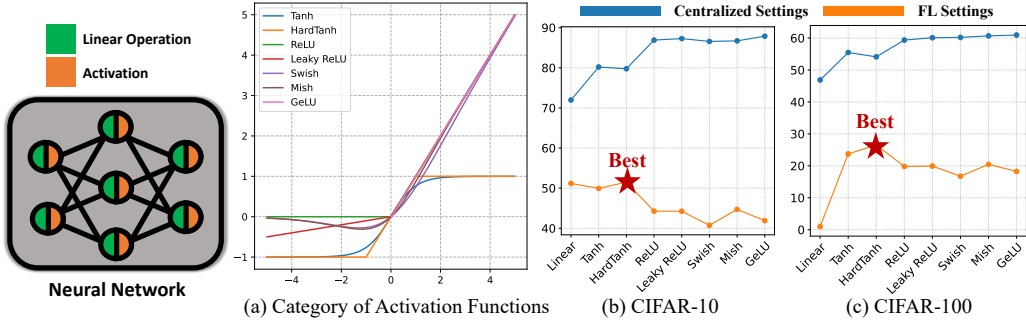

(a) Category of Activation Functions  (b) CIFAR-10  (c) CIFAR-100

Figure 1: (a) Plots of different activation functions. (b), (c) Accuracies on CIFAR-10 and CIFAR-100 according to the changes of activation functions, respectively. The blue line indicates the performances of models trained in a single central server. The orange line indicates those of models trained in FL environment where 20 clients participate in the training per round among the total 100 clients. We use a model with four convolution layers and one classifier. Here, 'Linear' indicates the model without any activation function. In one machine centralizing the training data, the more up-to-date activation function used, the better the performance (Blue line). In contrast, interestingly, HardTanh [7] prints the best server accuracy for both CIFAR-10 and CIFAR-100 under heterogeneous scenarios (Orange line). The detailed explanation about activation functions are provided in Appendix A.

Despite the popularity of FL, some options for federated model optimization remain under-explored. Designing FL-familiar training recipes is essential to optimizing model performance, but few studies have attempted to design a new recipe instead of using those intended for a centralized setting. Charles et al. [5] present an empirical analysis of the impact of hyperparameter settings for federated training dynamics from the perspective of a large cohort size. However, FL activation functions [37, 25, 34] have rarely been studied, whereas activation functions play a crucial role in facilitating generalization and convergence. We thus raise the seemingly doubtful question: *Do activation functions that are popular in centralized settings also produce good optima in FL?*

We conduct a pilot experiment to compare performance in centralized settings with performance in FL settings to answer this question. Figure 1 (b) and (c) show the accuracy of neural networks trained under a centralized setting and a FL setting according to the changes in activation functions. A shallow neural network that uses Tanh has better accuracy than ReLU, which is a silver bullet in the centralized deep learning field. The problems mentioned above lead us to the intriguing question:

*Do off-the-shelf activation functions intended for a centralized setting
also perform appropriately in the FL setting?*

In this work, we answer the question with thorough empirical evaluations: *the latest developed activation functions rather degrade the performance of the server as the heterogeneity gets severe.*

Several considerations (e.g., the total number of clients, client participation, heterogeneity) in selecting activation functions may improve significantly. Combining considerations may further boost the model accuracy. We experiment with various activation functions, including widely-used and rarely-used functions in the centralized setting, in various environments based on CIFAR-10 and CIFAR-100. The experiments identify an interesting phenomenon in FL in which applying activation functions like ReLU in stacked convolutional layers demonstrates low accuracy owing to the shape of the function. We also provide an analysis of the representation power according to the changes in activation functions for federated image classification. Our key contributions are summarized as follows:

- We provide guidelines for selecting activation functions in FL. FL has the following special considerations: number of clients, participation ratio, and non-IIDness. We provide guidelines for *cross-silo* settings (i.e., for a number of clients $\leq 20$) and *cross-device* settings (i.e., for a number of clients $\geq 100$); the suitable activation function depends on the situation.

- We provide an explanation for the performance degradetion (i.e., for the performance difference between centralized settings and FL settings) of activation functions, that are the preferred in a centralized setting. Specifically, we measure the similarities in the manner of weight parameters and representations.

- We empirically show that the HardTanh activation function [7] leads to a better optimum do other activation functions such as ReLU [39], Leaky ReLU [36], and GeLU [16] for severe

Table 1: Accuracy of centralized and FL settings in two datasets. Centralized settings train one server model using all training data and FL settings train 100 clients with heterogeneous data. We use $R = 0.2$, and $\alpha = 0.1$.

| Activation Function | CIFAR-10 | | CIFAR-100 | |
|---|---|---|---|---|
| | Centralized Setting | FL setting | Centralized Setting | FL Setting |
| Linear | 73.74±1.54 | 10.00 | 46.52±0.31 | 28.39 |
| Tanh | 81.00±0.89 | 52.58 | 55.39±0.24 | 30.75 |
| HardTanh | 80.64±0.77 | **54.43** | 54.31±0.46 | **31.76** |
| ReLU | 87.01±0.11 | 48.37 | 59.75±0.33 | 23.99 |
| Leaky ReLU | 87.30±0.03 | 48.34 | 60.28±0.22 | 24.04 |
| Swish | 86.50±0.08 | 46.16 | 60.02±0.47 | 21.55 |
| Mish | 86.38±0.29 | 50.02 | 60.67±0.03 | 24.98 |
| GeLU | **87.77±0.11** | 47.46 | **61.34±0.35** | 23.26 |

non-IID setting, a low participation rate, and with a large number of clients, and we provide benchmark data for activation functions in FL.

## 2 Related Work

This work is related to activation functions in neural networks, and FL methods. For details, refer to Appendix A.

## 3 Experiments

In this section, we compare several activation functions. We categorize the activation functions into two groups: (1) ReLU, Leaky ReLU, Swish, Mish, and GeLU as recent SOTA activation functions that are widely used in centralized settings; and (2) Tanh and HardTanh as Tanh-like activation functions that are not widely used in centralized settings.

### 3.1 Experimental Setup

**Dataset and Heterogenous Settings.**   We use two benchmark datasets: CIFAR-10 and CIFAR-100 [29]. We provide the descriptions of the datasets in Appendix B. To randomize the heterogeneous data, we assume that all client training data use class labels according to an independent categorical distribution for $N$ classes parameterized by the vector $q$:

$$q_i \geq 0, \ i \in [1, N] \quad \text{and} \quad \sum_{i \in [1,N]} q_i = 1$$

For the heterogeneous distribution, the *Dirichlet distribution* [18, 48], $q \sim \text{Dir}(\boldsymbol{\alpha})$ is used, where $\boldsymbol{\alpha}$ is an $N$-length concentration vector having all elements $\alpha > 0$, that is, the prior distribution for $N$ classes controls the heterogeneity of clients.

**Models.**   Our study focuses on compact models that are realistically possible in FL. Therefore, we use a simple `ConvNet` having four convolutional layers and one classifier; `ConvNet4` refers to `ConvNet` with four convolutional layers. The first convolution layer has 64 kernels, and deeper layers have a larger number of kernels [40]. For additional models, which have a shortcut and batch normalization layer, we use `Resnet20`, `Resnet32`, `Resnet44` [15], and `MobileNetv2` [43].

**Training Details.**   In this study, we conduct numerical experiments by varying the number of clients $N$, the client participation ratio $R$, and the Dirichlet distribution constant $\alpha$. We mainly exhibit the training of `ConvNet4` on heterogeneously distributed CIFAR-10 by modifying $R$ and $\alpha$. In the captions, we explain each $N$, $R$, and $\alpha$ value. The details are explained in Appendix B.

### 3.2 Comparative Experiments on the Changes in Activation Functions

Table 1 shows the result of both centralized and FL settings using CIFAR-10 and CIFAR-100 as the datasets. In centralized settings, GeLU shows the best performance, and other recent SOTA activation functions surpass the Tanh-like activation functions. However, in FL settings, the activation functions show a significantly different tendency. HardTanh achieves the highest accuracy. Furthermore, the recent SOTA activation functions show lower accuracy than Linear using CIFAR-100 as the dataset.

Table 2: Server accuracy of `ConvNet4` with four different Dirichlet constant values $\alpha$ (0.01, 0.1, 1, 10). We use $N = 100$ and $N = 20$ with $R = 0.2$.

| Activation Function | $N = 100$ | | | | $N = 20$ | | | |
|---|---|---|---|---|---|---|---|---|
| | $\alpha = 10$ | $\alpha = 1$ | $\alpha = 0.1$ | $\alpha = 0.01$ | $\alpha = 10$ | $\alpha = 1$ | $\alpha = 0.1$ | $\alpha = 0.01$ |
| Linear | 62.48 | 62.43 | 10.00 | 10.00 | 66.48 | 66.48 | 63.54 | 10.00 |
| Tanh | 64.49 | 64.14 | 52.58 | 29.50 | 70.22 | 69.55 | 65.97 | 27.64 |
| HardTanh | **65.27** | **65.40** | **54.43** | 30.09 | 70.53 | 70.01 | **66.53** | 28.92 |
| ReLU | 57.80 | 56.23 | 48.37 | 34.03 | 75.59 | 74.21 | 63.39 | 34.70 |
| Leaky ReLU | 57.85 | 56.16 | 48.34 | 33.92 | 75.36 | 74.23 | 63.67 | 35.15 |
| Swish | 52.62 | 51.48 | 46.16 | 35.65 | 72.58 | 71.58 | 64.57 | 36.93 |
| Mish | 57.30 | 55.08 | 50.02 | **38.94** | 73.69 | 72.95 | 66.47 | **37.89** |
| GeLU | 55.59 | 54.34 | 47.46 | 36.09 | **75.68** | **74.41** | 65.62 | 37.77 |

Table 3: Server accuracy of `ConvNet4` with four different participation ratios $R$ (0.1, 0.2, 0.3, 0.4). We use $N = 100$ and $N = 20$ with $\alpha = 0.1$.

| Activation Function | $N = 100$ | | | | $N = 20$ | | | |
|---|---|---|---|---|---|---|---|---|
| | $R = 0.4$ | $R = 0.3$ | $R = 0.2$ | $R = 0.1$ | $R = 0.4$ | $R = 0.3$ | $R = 0.2$ | $R = 0.1$ |
| Linear | 10.00 | 10.00 | 10.00 | 10.00 | 65.85 | 66.12 | 63.54 | 58.36 |
| Tanh | **62.61** | 61.79 | 52.58 | **46.61** | 69.22 | 67.75 | 65.97 | **62.67** |
| HardTanh | 59.45 | **61.75** | **54.43** | 41.90 | 69.25 | 68.80 | **66.53** | 43.96 |
| ReLU | 53.26 | 50.67 | 48.37 | 41.95 | 71.96 | 70.17 | 63.39 | 54.47 |
| Leaky ReLU | 53.17 | 50.76 | 48.34 | 42.05 | 66.12 | **70.22** | 63.67 | 54.46 |
| Swish | 51.72 | 49.45 | 46.16 | 40.00 | 70.36 | 67.99 | 64.57 | 53.74 |
| Mish | 56.67 | 52.80 | 50.02 | 43.37 | 71.30 | 69.34 | 66.47 | 60.38 |
| GeLU | 53.35 | 50.61 | 47.46 | 41.50 | **72.32** | 70.10 | 65.62 | 54.05 |

Activation functions have different accuracy drop; only recent SOTA activation functions have an accuracy drop near 40, while HardTanh and Tanh have 26.21 and 28.42 at CIFAR-10. As a result, we can find that the most popular activation function, ReLU (as well as recent SOTA activation functions), does not show outstanding performance in an FL setting.

### 3.3 Strategies for Selecting Activation Functions in FL

This section presents the experimental results and guidelines for selecting the activation functions for various FL settings. FL settings have various environmental limitations relative to centralized settings. It has additional components to consider, such as the number of clients, IID-ness, and the participation ratio.

**Number of Clients.** In different FL strategies, the number of clients varies. A *cross-silo* setting uses fewer than 20 clients, and a *cross-device* setting use more than 100 clients. As the number of clients differs, the drop in accuracy caused by the activation functions varies. The $\alpha = 0.1$ columns of Table 2 lists the accuracy for different client numbers. As the number of clients decreases, the overall drop in accuracy decreases. For both numbers of clients, HardTanh achieves the highest accuracy. However, the accuracy difference between the recent SOTA activation functions and Tanh-like activation functions is larger for 100 clients. Considering the observations for the number of clients, we hypothesize that as the number of clients increases, recent SOTA activation functions are increasingly affected and show a more significant accuracy drop.

**IID-ness.** With the Dirichlet distribution parameter $\alpha$, we can control the IID-ness: a larger value of $\alpha$ indicates higher IID-ness (lower heterogeneity). Table 2 presents the accuracy for different different values of $\alpha$. In most cases, HardTanh shows the highest accuracy. For 100 clients, at lower IID-ness the accuracy of the Tanh-like activation functions surpasses the accuracy of the recent SOTA activation functions with larger accuracy gap. For 20 clients, the recent SOTA activation functions surpass the Tanh-like activation functions at high IID-ness. Activation's shape cause severe accuracy drop of the recent SOTA activations with low IID-ness, which we discuss at section 4. The low accuracy of the Tanh-like activation functions at $\alpha = 0.01$ occurs due to tough training settings and the Tanh-like activation functions fail to find the optimum, such as Linear.

**Participation Ratio.** The participation of clients is limited in FL depending on the environment. In a *cross-silo* setting, high participation may be possible, whereas only limited participation is available in a the *cross-device* setting. Table 3 shows the accuracy in FL settings for four different values of the

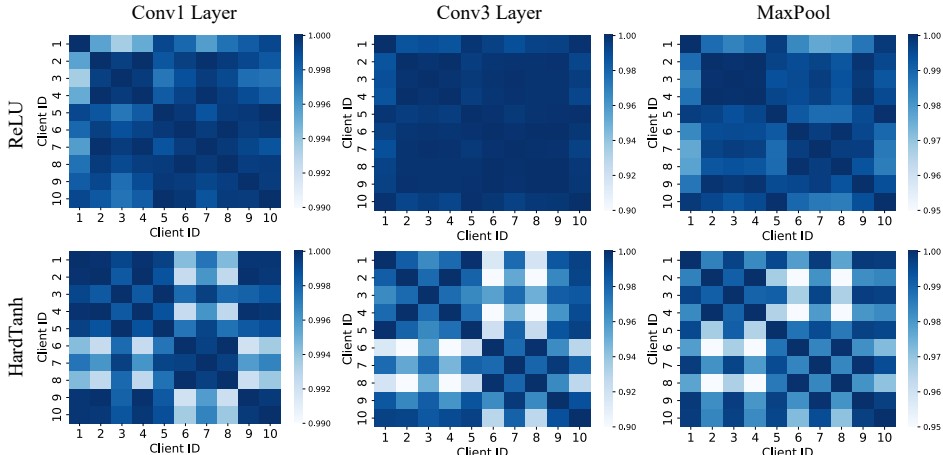

Figure 2: CKA similarity between 10 client using test images of CIFAR-10. Each client's model is the model before $100th$ aggregation. We use $N = 100$ with $R = 1.0$ and $\alpha = 0.1$ for training. We calculate the CKA similarity using features passing through each layer and its activation function.

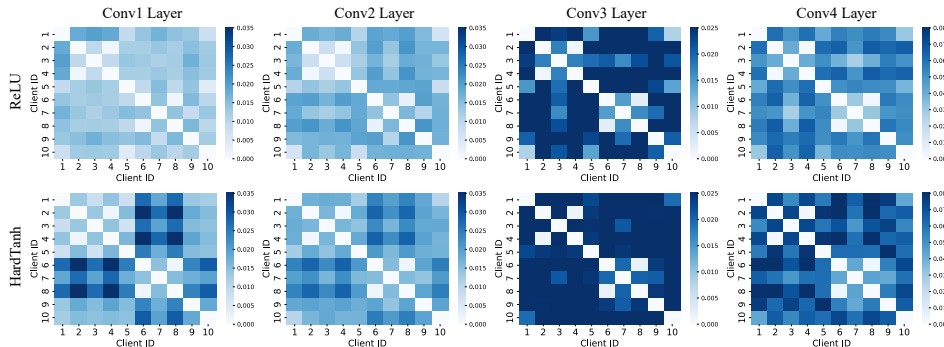

Figure 3: Weight difference between 10 client selected in Figure 2. We calculate the weight difference by subtracting each client's weight from the same layer and normalizing it with $L_2$norm.

participation ratio. For 100 clients, the Tanh-like activation functions achieve the highest accuracy. With 20 clients, however, there is a noticeably larger accuracy drop as participation decreases for the most recent SOTA activation functions. With higher client participation, *client drift* [25, 26, 42] reduces, and the influence of the activation function's shape of the recent SOTA activation functions drops and occur smaller accuracy.

Different FL settings components affect accuracy when different activation functions are used, according to the observations above. The number of clients is the most dominant component, and it interacts with the influence of other components when a small number of clients is used. Therefore, the Tanh-like activation functions are favorable for a large number of clients, such as in a *cross-device* setting. For a *cross-silo* setting, with low data IID-ness, and a low ratio of client participation the Tanh-like activation functions are preferred. Conversely, the recent SOTA activation functions are preferred for a *cross-silo* setting, with high data IID-ness, and high ratio of client participation.

### 3.4 Additional Experiment

Additional experiments with different FL method and models showed a similar tendency, where Tanh-like activation functions surpassed recent SOTA activation functions as observed above. We chose FedProx as the additional FL method and `Resnet20`, `Resnet32`, `Resnet44`, and `MobileNetv2` as the additional models. For details of the experimental results, refer to Appendix C.

## 4 Analysis

We present thorough investigations on model behavior and the changes in representation during the local training to answer this question: *do recent SOTA activation functions have a disadvantage in FL settings?*

### 4.1 Investigation of Weight Parameters and Latent Representations

The activation function selects important features to pass through each layer, and the model is trained using these features. The number of selected features varies according to the shape of an activation function. In a conventional centralized setting, a single model can access all the data and select optimal training features. However, in an FL setting, each client can only access a portion of the data, which is partitioned in the non-IID condition, and each client trains its model to select features that are important to itself. This results in a phenomenon known as *client drift*. During the FL aggregation step, a problem arises where important features for the global optimum cannot be selected due to client drift. This phenomenon appears to be severe when the recent SOTA activation functions are used. Due to the shape of their activation functions, the excluded features are greater in number than for Tanh-like activation functions, and a severe accuracy drop occurs. Figure 5 (a) in Appendix C shows the feature distribution after the first convolution layer. The recent SOTA activation functions exclude more feature and have a high density near 0. The empirical results in section 3 show that for a large number of clients and a small client participation ratio, the client drift increases, and the accuracy drop for recent SOTA activation functions is maximal.

This can be simply summarized by saying that, the number of excluded features varies due to the different shapes of the activation functions, and as an activation function excludes more feature values, the drop in accuracy increases. Recent SOTA activation functions tend to exclude features smaller than 0. Tanh-like activation functions, on the other hand, exclude features that are smaller than -1 and larger than 1. Tanh-like activation functions have low sensitivity about the accuracy drop in the FL aggregation step because they exclude a much smaller number of features than do recent SOTA activation functions. Additionally, HardTanh and Tanh show better accuracy than Linear in most situations due to their existence of non-linearity. Figure 5 (b) in Appendix C shows the accuracy in FL settings for different versions of `ConvNet`, which have different numbers of convolutional layers. Deeper models exclude more features, and as a result, `ConvNet` used with recent SOTA activation functions shows a significant drop in accuracy.

To observe how the activation function affects `ConvNet4`, we perform two additional experimental studies on heterogeneous local models. For simplicity, we look at 10 clients out of a total 100. First, we perform Centered Kernel Alignment (CKA) [28] to measure the similarity of the output features between different clients. Each client trains the server model for 5 epochs with their own non-IID data. In Figure 2, ReLU has a higher CKA similarity than HardTanh in every layer. Because feature selection fails for the server model, as indicated above, ConvNet4 with ReLU has a smaller feature change than does HardTanh, which indicates a lower learning ability. Second, we calculate the weight differential to check how similar other clients weights are. Figure 3 reveals that ConvNet4 with ReLU has a smaller weight difference in each layer than does HardTanh, which also indicates ReLU has a limited learning ability.

### 4.2 Analysis of Landscape

To check if each activation functions reaches the global optimum, we visualize the 2-D landscape of Tanh, HardTanh, ReLU, and Leaky ReLU in Appendix D. Figure 5 shows the 2-D landscape of each activation function. It seems that ReLU and Leaky ReLU fails to approach to the global optimum. In contrast, Tanh and HardTanh succeed in reaching the global optimum. Furthermore, ReLU and Leaky ReLU has steep slope in their landscape where Tanh and HardTanh does not.

These findings suggest that the sensitivity of the activation functions in the FL aggregation step, where the accuracy drop occurs, is indicated by the shape of the activation functions. The accuracy drop increases as the number of features excluded by an activation function increases, and it reaches a maximum with severe client drift; a large number of clients, a low client participation ratio, and high heterogeneity.

## 5 Conclusion

This study clarifies that the drop in accuracy varies according to the activation function in FL. Our key finding is that the accuracy of the recent SOTA activation functions drops in an FL setting due to the shape of the functions, and HardTanh outperforms other activation functions in most environments. Additionally, we provide guidelines and benchmark data for selecting activation functions in various FL settings. We believe that further research is needed to find the silver bullet of activation function in an FL setting, which our paper did not address, and that our work will inspire others.

## Acknowledgement

This work was supported by Institute of Information & communications Technology Planning & Evaluation (IITP) grant funded by Korea government (MSIT) [No. 2021-0-00907, Development of Adaptive and Lightweight Edge-Collaborative Analysis Technology for Enabling Proactively Immediate Response and Rapid Learning, 90%] and [No. 2019-0-00075, Artificial Intelligence Graduate School Program (KAIST), 10%].

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

# Appendix

## A  Related Work

### A.1  Activation Functions in Neural Networks

In deep neural networks, using activation functions is a ubiquitous technique for learning non-linear latent representations; an input signal is transformed into the non-linear output centered on zero. Recent evolution occurs along with the enhancement of representation power and the efficiency of computational costs. In the experiment, we use the following non-linear activation functions:

**Tanh.**  The equation of Tanh is:

$$\tanh(x) = \frac{e^x - e^{-x}}{e^x + e^{-x}}$$

It is known that it is zero-centered, but computationally expensive and causes vanishing gradient problems as neural networks become deeper.

**HardTanh.**  It is another variant of the hyperbolic tangent function, which represents computationally more efficient form of tanh:

$$\text{HTanh}(x) = \min(1, \max(-1, x))$$

**ReLU.**  The equation of ReLU [12, 20, 39] is:

$$\text{ReLU}(x) = \max(0, x)$$

ReLU is the abbreviation of *Rectified Linear Unit*, a modified linear function. When the input value of ReLU is negative, the gradient of its output value is zero; the model does not learn. ReLU has shown great performance with Convolutional Neural Networks (CNN) [30, 31]. Since ReLU is computationally cheap, it is still commonly used regardless of numerous attempts to replace it [36, 11, 14, 6, 27, 8].

**Leaky ReLU.**  The equation of Leaky ReLU [36] is:

$$\text{LReLU}(x) = \max(0.01x, x)$$

This function was created to solve the *Dying ReLU* problem in ReLU. It is ReLU multiplied by a tiny constant on the negative part. Due to the small range, the graphs are drawn almost similarly with ReLU.

**Swish.**  The equation of Swish [41] is:

$$\text{Swish}(x) = x \cdot sigmoid(x)$$

This function shows better accuracy than ReLU in deep neural networks regardless of batch size.

**Mish.**  The equation of Mish [38] is:

$$\text{Mish}(x) = x \cdot tanh(ln(1 + e^x)))$$

This function has the characteristic of allowing gradients to flow better than the Relu Zero Bound because it allows some negative numbers.

**GeLU.**  The equation of GeLU [16] is:

$$\text{GeLU}(x) = x \cdot \frac{1}{2}[1 + tanh[\sqrt{2\pi}(x + 0.044715x^3)]]$$

This function is widespread on BERT, GPT, VIT models. GeLU is derived by combining the characteristics of dropout, zoneout, and ReLU.
Activation functions have been devised for the *gradient exploding/vanishing* issues; the magnitude of gradients becomes either near zero or infinite during backward propagation. A general choice for activation functions is ReLU which raises significantly cheap computational costs. Still, it is not differentiable at zero as well as causes significant dying neurons forgetting the information during propagation. Recently, some works [41, 38] lead to more smooth optima by designing self-regularized gradients. On the other hand, these studies are based on centralized settings, whereas those based on FL settings are badlands. Details of activation functions are explained in Appendix B.

## A.2 Federated Learning Methods

Federated optimization methods manage to handle multiple clients without collecting data, and they use server weights from a central server to coordinate the global model across the network. In particular, these methods aim to minimize the following objective function:

$$\min_{w} f(w) \quad \text{where} \quad f(w) = \frac{1}{N} \sum_{k=1}^{N} f^{(k)}(w) \tag{1}$$

where $f^{(k)}$ is the loss function based on the client $k$. $N$ is the total number of clients. At each round $K \ll N$ clients are selected from the total devices. The selected clients run each local model using SGD for $E$ number of local epochs and finally aggregate the selected models at the server model.

In the FL environment, the oracle of a global model can be drifted by optimizing the local clients because the statistical data heterogeneity causes different local optimums widely apart from each other. It is called *Client drift* [25, 26, 42] which indicates the inconsistency among each optimum. Recently, some works prevent *Client drift* designing aggregation methods; Wang et al. [47] present a method of normalized averaging that removes objective inconsistency, and Zhang et al. [49] propose a training algorithm for group knowledge transfer, which allows each client to keep a personalized prediction on the server to assist the local training of others.

**Federated Averaging (FedAvg [37])** uses the local server for stochastic gradient descent (SGD) locally for $E$ number of epochs. As a result, the selected client resulting $k$'s weight is updated as $w_k$. At each round, to aggregate the local client models, FedAvg sums and averages for the server model parameters formulated as:

$$w^t = \frac{1}{K} \sum_{k \in S_t} w_k^t \tag{2}$$

where $w^t$ is the server weight of the $t$-th round, $w_i^t$ is the client $k$'s model after local training using $w^{t-1}$, and $S_t$ is the client set. McMahan et al. [37] empirically shows the significance of tuning the hyperparameters in FL training additionally, we present that with respect to the architectural and operational side.

## A.3 Algorithms of Federated Learning Methods

We use both FedAvg and FedProx for federated learning methods. Algorithm 1 shows the algorithm of FedAvg and Algorithm 2 shows the algorithm of FedProx. FedProx is similar to FedAvg in that it selects a selection of clients at each round, performs local training, and then averages client's weight to generate a global update. However, the difference between FedAvg and FedProx is shown in line 6. For local training, FedAvg trains each client's model using SGD with its local data whereas FedProx, trains each client with additional proximal term, $\frac{\mu}{2} \|w - w^t\|^2$. Using the proximal term which contributes to the method's stability by efficiently reducing the impact of variable modifications.

---

**Algorithm 1** Federated Averaging (FedAvg)

---

1: **Input:** $K, T, \eta, E, w^0, N, k \in [1, \cdots, N]$
2: **for** $t = 0, \cdots, T-1$ **do**
3:     Server selects a subset $S_t$ randomly which includes number of $K$ devices
4:     Server send $w^t$ to all selected devices
5:     **for** $i = 0, \cdots, E-1$ **do**
6:         Selected device $k \in S_t$ updates their local weight $w_k^{t+1}$ using SGD with step-size $\eta$
7:     **end for**
8:     Selected device $k \in S_t$ sends their local weight $w_k^{t+1}$ back to the server
9:     Server aggregates the local weights, $w_k^{t+1}$, and gets new server weight $w^{t+1} = \frac{1}{K} \sum_{k \in S_t} w_k^t$
10: **end for**

---

**Algorithm 2** FedProx

1: **Input:** $K, T, \eta, \mu, E, w^0, N, k \in [1, \cdots, N]$
2: **for** $t = 0, \cdots, T - 1$ **do**
3:     Server selects a subset $S_t$ randomly which includes number of $K$ devices
4:     Server send $w^t$ to all selected devices
5:     **for** $i = 0, \cdots, E - 1$ **do**
6:         Selected device $k \in S_t$ updates their local weight $w_k^{t+1} \approx \min_w h^{(k)}(w; w^t) = f^{(k)}(w) + \frac{\mu}{2} \|w - w^t\|^2$ with step-size $\eta$
7:     **end for**
8:     Selected device $k \in S_t$ sends their local weight $w_k^{t+1}$ back to the server
9:     Server aggregates the local weights, $w_k^{t+1}$, and gets new server weight $w^{t+1} = \frac{1}{K} \sum_{k \in S_t} w_k^t$
10: **end for**

## B  Implementation Details

### B.1  Model Architecture

Figure 4 shows `ConvNet` with five different depth (3,4,5,6, and 7). Each version of `ConvNet` has convolution layer corresponding to the number after the model (e.g., `ConvNet3` has three convolution layers and `ConvNet7` has seven convolution layers). The details of each convolution layer is shown in Table 4. `ConvNet` with different depth use convolution layers sequentially from the top in Table 4 (e.g., `ConvNet3` use Conv1, Conv2, Conv3 whereas, `ConvNet7` use Conv1, Conv2, Conv3, Conv4, Conv5, Conv6, and Conv7).

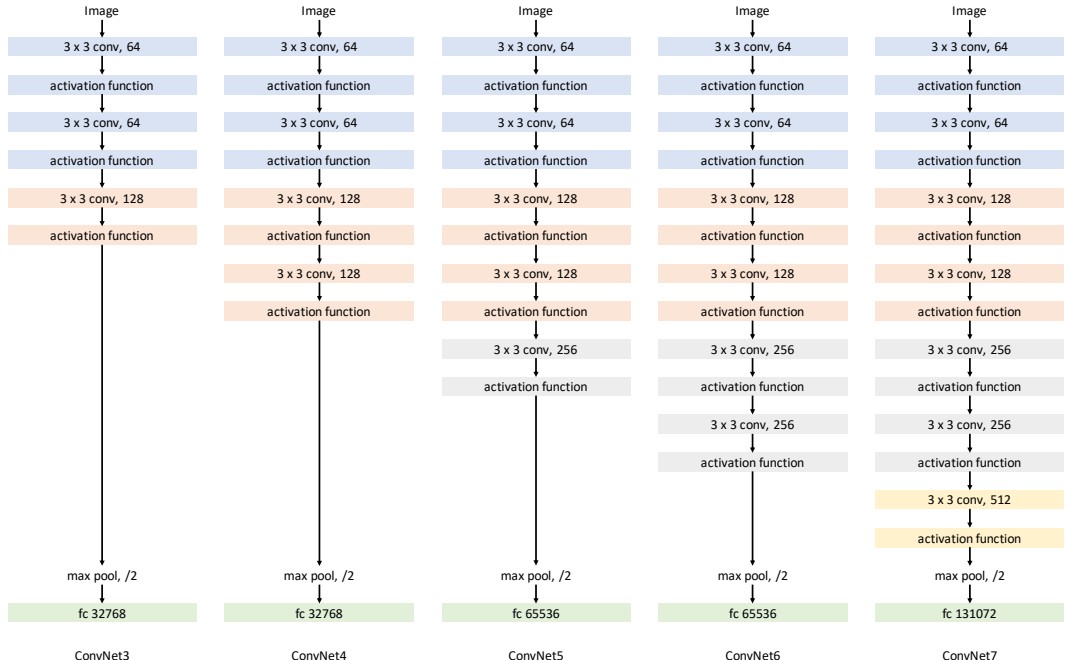

Figure 4: The architecture of `ConvNet` with different number of convolution layers.

### B.2  Dataset statistics

We use both CIFAR-10 and CIFAR-100 for our experiments. As shown in Table 5, CIFAR-10 consists of 60000 images of size $32 \times 32$. It is divided into 10 classes, and each class consists of 6000 images. Also, 5000 are training data and 1000 are test data. CIFAR-100 also consists of 60000 images with a

Table 4: Setting of each convolution layer in `ConvNet`. Each version of `ConvNet` uses the number convolution layer as much as their version number. (i.e `ConvNet3` uses conv1 through conv2 and `ConvNet7` uses conv1 through conv7.)

| Layer | Number of Input Filter | Number of Output Filter | Kernel Size | Padding |
|-------|-----------------------|-------------------------|-------------|---------|
| Conv1 | 3 | 64 | 3×3 | 1 |
| Conv2 | 64 | 64 | 3×3 | 1 |
| Conv3 | 64 | 128 | 3×3 | 1 |
| Conv4 | 128 | 128 | 3×3 | 1 |
| Conv5 | 128 | 256 | 3×3 | 1 |
| Conv6 | 256 | 256 | 3×3 | 1 |
| Conv7 | 256 | 512 | 3×3 | 1 |

size of 32×43. It is classified into 100 classes, and each class consists of 600 images. These 100 classes are divided into 20 superclasses, which we do not use in our experiments. Also, 500 are training data and 100 are test data.

Table 5: Data set statistics.

| Data set | Train examples | Test examples | Class Number | Task |
|----------|----------------|---------------|--------------|------|
| CIFAR-10 | 50,000 | 10,000 | 10 | Image Classification |
| CIFAR-100 | 50,000 | 10,000 | 100 | Image Classification |

## B.3   Training Details

In this study, we conduct numerical experiments by changing the number of clients $N$, the client participation ratio $R$, and the Dirichlet distribution constant $\alpha$. We adapt FedAvg and perform 200 rounds with 5 local epochs using a learning rate of 0.01, with a learning decay of 0.1 at the 50th and 75th round, a weight decay of $1e-4$, and a momentum of 0.9. Limited client numbers are available in different FL settings; in a *cross-silo* setting, a small number of clients are available, and a large number of clients are requested in a *cross-device*. For the *cross-silo* setting, we use $N = 20$ and $R = 0.2$ in which we select 4 clients each round. For the *cross-device* setting, we use $N = 100$ and $R = 0.2$, which we select 20 clients each round.

We mainly exhibit the training of `ConvNet4` on CIFAR-10 heterogeneously distributed by modifying the $\alpha$ in the Dirichlet distribution. In the captions, we explain each $N$, $R$, and $\alpha$ value.

## C   Additional Experiment Result

### C.1   ConvNet4 Result

Experiment results in section 3 fix variables such as $N$, $R$, and $\alpha$. Table 6 shows the result with all combinations of $R$ and $\alpha$ with $N = 100$. Table 7 shows the result using CIFAR-100 with all combinations of $R$ and $\alpha$ with $N = 100$. We implement all experiments on Single GPU NVIDIA 2080-Ti.

Figure 5 (a) shows the feature distribution in `ConvNet4` with different activation functions after passing the first convolution layer. After passing the convolution layer and activation function, we flatten the feature values and draw a distribution. The feature densities of the recent SOTA activation functions have discreteness and a high density near 0 due to their function shape. As the features pass deeper layers, the densities near 0 of the recent SOTA activation functions increase compared with the Tanh-like activation functions. In a centralized setting, the recent SOTA activation functions, especially ReLU work well due to its discreteness at 0 and its function shape, which is similar to a linear function which prevents the vanishing gradient and exploding gradient problems [19].

Table 6: Server accuracy of `ConvNet4` using four different $\alpha$ and four different participation $R$, where $N = 100$. The most right columns show the accuracy drop as non-IIDness increases. Since using Linear at $\alpha = 0.01, 0.1$ with participation ratio 0.1, 0.2, 0.3, and 0.4 could not train, leave blank at $\alpha = 0.01 \rightarrow 0.1, \alpha = 1 \rightarrow \alpha = 0.1$ with participation ratio 0.1, 0.2, 0.3, and 0.4.

| Participation Ratio | Activation Function | $\alpha = 10$ | $\alpha = 1$ | $\alpha = 0.1$ | $\alpha = 0.01$ | $\alpha = 10 \rightarrow 1$ | $\alpha = 1 \rightarrow 0.1$ | $\alpha = 0.1 \rightarrow 0.01$ |
|---|---|---|---|---|---|---|---|---|
| | Linear | 58.24 | 57.38 | 10.00 | 10.00 | 0.86 | - | - |
| | Tanh | 58.98 | 57.86 | **46.61** | 31.97 | 1.12 | 11.25 | 14.64 |
| | HardTanh | **59.85** | **59.62** | 41.90 | 30.34 | 0.23 | 17.72 | 11.56 |
| 0.1 | ReLU | 51.61 | 49.73 | 41.95 | 28.83 | 1.88 | 7.78 | 13.12 |
| | Leaky ReLU | 51.60 | 49.76 | 42.05 | 28.90 | 1.84 | 7.71 | 13.15 |
| | Swish | 47.01 | 46.28 | 40.00 | 29.88 | 0.73 | 6.28 | 10.12 |
| | Mish | 50.00 | 49.59 | 43.37 | **33.13** | 0.41 | 6.22 | 10.24 |
| | GeLU | 48.64 | 48.02 | 41.50 | 31.61 | 0.62 | 6.52 | 9.89 |
| | Linear | 62.48 | 62.43 | 10.00 | 10.00 | 0.05 | - | - |
| | Tanh | 64.49 | 64.14 | 52.58 | 29.50 | 0.35 | 11.56 | 23.08 |
| | HardTanh | **65.27** | **65.40** | **54.43** | 30.09 | -0.13 | 10.97 | 24.34 |
| 0.2 | ReLU | 57.80 | 56.23 | 48.37 | 34.03 | 1.57 | 7.86 | 14.34 |
| | Leaky ReLU | 57.85 | 56.16 | 48.34 | 33.92 | 1.69 | 7.82 | 14.42 |
| | Swish | 52.62 | 51.48 | 46.16 | 35.65 | 1.14 | 5.32 | 10.51 |
| | Mish | 57.30 | 55.08 | 50.02 | **38.94** | 2.22 | 5.06 | 11.08 |
| | GeLU | 55.59 | 54.34 | 47.46 | 36.09 | 1.25 | 6.91 | 11.37 |
| | Linear | 64.66 | 64.46 | 10.00 | 10.00 | 0.20 | - | - |
| | Tanh | 67.34 | 66.57 | **61.79** | 31.35 | 0.77 | 4.78 | 30.44 |
| | HardTanh | **68.35** | **67.86** | 61.75 | 29.93 | 0.49 | 6.11 | 31.82 |
| 0.3 | ReLU | 62.43 | 60.38 | 50.67 | 37.91 | 2.05 | 9.71 | 12.76 |
| | Leaky ReLU | 62.62 | 60.34 | 50.76 | 38.02 | 2.28 | 9.58 | 12.74 |
| | Swish | 58.04 | 55.88 | 49.45 | 39.78 | 2.16 | 6.43 | 9.67 |
| | Mish | 65.53 | 62.21 | 52.80 | **42.57** | 3.32 | 9.41 | 10.23 |
| | GeLU | 62.39 | 59.20 | 50.61 | 39.14 | 3.19 | 8.59 | 11.47 |
| | Linear | 65.71 | 65.83 | 10.00 | 10.00 | -0.12 | - | - |
| | Tanh | 68.83 | 68.37 | **62.61** | 29.49 | 0.46 | 5.76 | 33.12 |
| | HardTanh | **69.95** | **69.35** | 59.45 | 27.22 | 0.60 | 9.90 | 32.23 |
| 0.4 | ReLU | 66.88 | 64.39 | 53.26 | 40.96 | 2.49 | 11.13 | 12.30 |
| | Leaky ReLU | 67.02 | 64.60 | 53.17 | 41.17 | 2.42 | 11.43 | 12.00 |
| | Swish | 64.78 | 60.13 | 51.72 | 41.86 | 4.65 | 8.41 | 9.86 |
| | Mish | 68.96 | 67.27 | 56.67 | **44.41** | 1.69 | 10.60 | 12.26 |
| | GeLU | 67.59 | 63.42 | 53.35 | 40.99 | 4.17 | 10.07 | 12.36 |

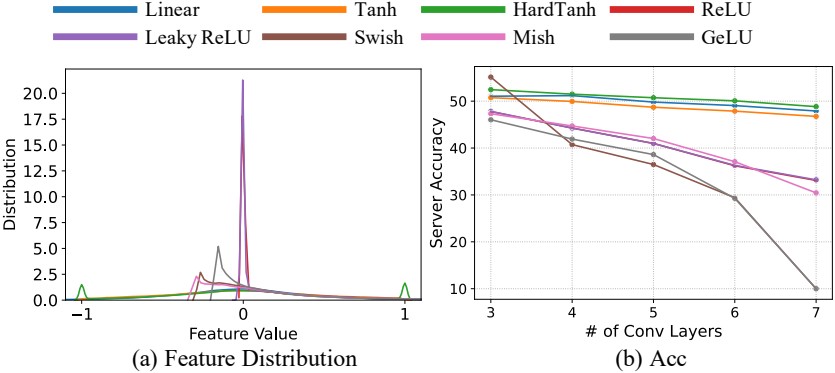

(a) Feature Distribution         (b) Acc

Figure 5: (a) shows the distribution of the feature of all images of class 0 in the CIFAR-10 test dataset passing through the first convolution layer and its activation function. (b) shows the accuracy of `ConvNet` in Federated Setting with Different widths and depths. For both figure, we use $N = 100$, $R = 0.2$ and $\alpha = 0.1$.

Table 7: Server accuracy of `ConvNet4` using CIFAR-100 as dataset and $N = 100$. The most right columns show the accuracy drop as non-IIDness increases. Since using Linear at $\alpha = 0.01$ with participation ratio 0.1, 0.2, 0.3, and 0.4 could not train, leave blank at $\alpha = 0.01 \rightarrow 0.1$ with participation ratio 0.1, 0.2, 0.3, and 0.4.

| Participation Ratio | Activation Function | $\alpha = 10$ | $\alpha = 1$ | $\alpha = 0.1$ | $\alpha = 0.01$ | $\alpha = 10 \rightarrow 1$ | $\alpha = 1 \rightarrow 0.1$ | $\alpha = 0.1 \rightarrow 0.01$ |
|---|---|---|---|---|---|---|---|---|
| 0.1 | Linear | 25.94 | 25.57 | 22.58 | 1.00 | 0.37 | 2.99 | - |
| | Tanh | 26.56 | 26.03 | 21.81 | 13.47 | 0.53 | 4.22 | 8.34 |
| | HardTanh | **29.04** | **28.66** | **23.70** | **14.66** | 0.38 | 4.96 | 9.04 |
| | ReLU | 21.83 | 22.67 | 17.98 | 10.97 | -0.84 | 4.69 | 7.01 |
| | Leaky ReLU | 21.95 | 22.77 | 18.12 | 11.13 | -0.82 | 4.65 | 6.99 |
| | Swish | 19.66 | 20.67 | 15.41 | 11.10 | -1.01 | 5.26 | 4.31 |
| | Mish | 23.75 | 23.30 | 18.42 | 11.86 | 0.45 | 4.88 | 6.56 |
| | GeLU | 21.21 | 21.77 | 16.65 | 11.15 | -0.56 | 5.12 | 5.50 |
| 0.2 | Linear | 31.78 | 31.89 | 28.39 | 1.00 | -0.11 | 3.50 | - |
| | Tanh | 34.01 | 34.74 | 30.75 | 19.56 | -0.73 | 3.99 | 11.19 |
| | HardTanh | **35.68** | **36.08** | **31.76** | **21.93** | -0.40 | 4.32 | 9.83 |
| | ReLU | 26.91 | 27.59 | 23.99 | 15.61 | -0.68 | 3.60 | 8.38 |
| | Leaky ReLU | 26.93 | 27.77 | 24.04 | 15.68 | -0.84 | 3.73 | 8.36 |
| | Swish | 26.39 | 26.14 | 21.55 | 14.57 | 0.25 | 4.59 | 6.98 |
| | Mish | 29.40 | 28.89 | 24.89 | 16.41 | 0.51 | 4.00 | 8.48 |
| | GeLU | 27.66 | 26.73 | 23.26 | 15.08 | 0.93 | 3.47 | 8.18 |
| 0.3 | Linear | 35.50 | 35.32 | 31.47 | 1.00 | 0.18 | 3.85 | - |
| | Tanh | 37.68 | 38.76 | 34.24 | 23.11 | -1.08 | 4.52 | 11.13 |
| | HardTanh | **39.05** | **39.40** | **35.03** | **24.09** | -0.35 | 4.37 | 10.94 |
| | ReLU | 29.81 | 30.24 | 27.31 | 17.92 | -0.43 | 2.93 | 9.39 |
| | Leaky ReLU | 29.74 | 30.26 | 27.45 | 17.97 | -0.52 | 2.81 | 9.48 |
| | Swish | 29.96 | 29.53 | 25.34 | 17.37 | 0.43 | 4.19 | 7.97 |
| | Mish | 32.33 | 32.71 | 29.19 | 20.16 | -0.38 | 3.52 | 9.03 |
| | GeLU | 30.24 | 29.77 | 27.24 | 17.89 | 0.47 | 2.53 | 9.35 |
| 0.4 | Linear | 37.48 | 37.53 | 33.79 | 1.00 | -0.05 | 3.74 | - |
| | Tanh | 40.01 | 40.85 | 37.13 | 26.35 | -0.84 | 3.72 | 10.78 |
| | HardTanh | **41.69** | **41.73** | **37.41** | **27.44** | -0.04 | 4.32 | 9.97 |
| | ReLU | 30.75 | 31.45 | 30.55 | 20.51 | -0.70 | 0.90 | 10.04 |
| | Leaky ReLU | 31.05 | 31.61 | 30.79 | 20.61 | -0.56 | 0.82 | 10.18 |
| | Swish | 32.54 | 32.37 | 27.61 | 19.45 | 0.17 | 4.76 | 8.16 |
| | Mish | 34.64 | 35.94 | 33.11 | 22.01 | -1.30 | 2.83 | 11.10 |
| | GeLU | 32.37 | 32.12 | 29.27 | 20.73 | 0.25 | 2.85 | 8.54 |

## C.2 Resnet Result

Table 8, Table 9, and Table 10 shows the result of `Resnet20`, `Resnet32`, and `Resnet44` with all combinations of $R$ and $\alpha$ with $N = 100$. For all values of $R$ and $\alpha$, `Resnet20` using HardTanh shows the highest accuracy. As model gets deeper, `Resnet32` and `Resnet44`, in some cases recent SOTA activation functions overcome Tanh-like activation functions. It seems to occur due to shortcut's existence. A deeper layer can use features that the activation function has excluded via a shortcut [15], which helps to prevent the recent SOTA activation functions' accuracy drop.

Table 8: Server accuracy of `Resnet20` using four different $\alpha$ and four different participation $R$, where $N = 100$. The most right columns show the accuracy drop as non-IIDness increases.

| Participation Ratio | Activation Function | $\alpha = 10$ | $\alpha = 1$ | $\alpha = 0.1$ | $\alpha = 0.01$ | $\alpha = 10 \to 1$ | $\alpha = 1 \to 0.1$ | $\alpha = 0.1 \to 0.01$ |
|---|---|---|---|---|---|---|---|---|
| 0.1 | Linear | 49.01 | 48.60 | 38.60 | 24.63 | 0.41 | 10.00 | 13.97 |
| | Tanh | 50.68 | 49.20 | 38.83 | 23.82 | 1.48 | 10.37 | 15.01 |
| | HardTanh | **51.23** | **49.44** | **39.28** | **23.91** | 1.79 | 10.16 | 15.37 |
| | ReLU | 48.81 | 48.12 | 37.53 | 23.34 | 0.69 | 10.59 | 14.19 |
| | Leaky ReLU | 48.63 | 48.15 | 37.71 | 23.50 | 0.48 | 10.44 | 14.21 |
| 0.2 | Linear | 56.93 | 56.96 | 45.73 | 26.70 | -0.03 | 11.23 | 19.03 |
| | Tanh | **59.66** | 57.42 | **46.58** | 26.50 | 2.24 | 10.84 | 20.08 |
| | HardTanh | 59.60 | **57.54** | 46.19 | **26.52** | 2.06 | 11.35 | 19.67 |
| | ReLU | 56.90 | 56.95 | 45.35 | 25.73 | -0.05 | 11.60 | 19.62 |
| | Leaky ReLU | 56.78 | 56.49 | 45.18 | 26.10 | 0.29 | 11.31 | 19.08 |
| 0.3 | Linear | 63.17 | 61.67 | 49.35 | 28.25 | 1.50 | 12.32 | 21.07 |
| | Tanh | **64.63** | **61.74** | 49.69 | **27.90** | 2.89 | 12.05 | 21.79 |
| | HardTanh | 64.41 | 61.59 | **50.16** | 27.86 | 2.82 | 11.43 | 22.30 |
| | ReLU | 61.99 | 61.59 | 48.71 | 27.58 | 0.40 | 12.88 | 21.13 |
| | Leaky ReLU | 61.87 | 61.34 | 48.38 | 27.38 | 0.53 | 12.96 | 21.00 |
| 0.4 | Linear | 66.24 | 65.30 | 52.09 | 30.17 | 0.94 | 13.21 | 21.92 |
| | Tanh | **68.08** | **65.05** | 52.92 | 28.92 | 3.03 | 12.13 | 24.00 |
| | HardTanh | 67.90 | 64.91 | **53.69** | **29.51** | 2.99 | 11.22 | 24.18 |
| | ReLU | 66.10 | 64.94 | 52.54 | 28.28 | 1.16 | 12.40 | 26.26 |
| | Leaky ReLU | 66.04 | 64.82 | 52.23 | 28.70 | 1.22 | 12.59 | 23.53 |

Table 9: Server accuracy of `Resnet32` using four different $\alpha$ and four different participation $R$, where $N = 100$. The most right columns show the accuracy drop as non-IIDness increases.

| Participation Ratio | Activation Function | $\alpha = 10$ | $\alpha = 1$ | $\alpha = 0.1$ | $\alpha = 0.01$ | $\alpha = 10 \to 1$ | $\alpha = 1 \to 0.1$ | $\alpha = 0.1 \to 0.01$ |
|---|---|---|---|---|---|---|---|---|
| 0.1 | Linear | 50.06 | 47.94 | 37.39 | 21.95 | 2.12 | 10.55 | 15.44 |
| | Tanh | 49.55 | 47.15 | 38.24 | 21.61 | 2.40 | 8.91 | 16.63 |
| | HardTanh | 50.04 | 47.99 | **38.34** | 22.37 | 2.05 | 9.65 | 15.97 |
| | ReLU | 50.68 | 48.39 | 37.87 | **24.29** | 2.29 | 10.52 | 13.58 |
| | Leaky ReLU | **50.71** | **48.57** | 37.33 | 23.91 | 2.14 | 11.24 | 13.42 |
| 0.2 | Linear | 60.12 | 56.84 | 43.01 | 24.98 | 3.28 | 13.83 | 18.03 |
| | Tanh | **60.03** | 56.81 | 43.92 | 25.90 | 3.22 | 12.89 | 18.02 |
| | HardTanh | 59.30 | **57.39** | **43.99** | 25.51 | 1.91 | 13.40 | 18.48 |
| | ReLU | 58.52 | 56.34 | 43.09 | **26.66** | 2.18 | 13.25 | 16.43 |
| | Leaky ReLU | 58.97 | 56.16 | 43.42 | 26.42 | 2.81 | 12.74 | 17.00 |
| 0.3 | Linear | 64.51 | 62.20 | 46.15 | 28.63 | 2.31 | 16.05 | 17.52 |
| | Tanh | **64.60** | 61.96 | **47.77** | 28.04 | 2.64 | 14.19 | 19.73 |
| | HardTanh | 63.74 | **62.20** | 47.73 | 28.03 | 1.54 | 14.47 | 19.70 |
| | ReLU | 64.23 | 61.58 | 46.49 | **28.90** | 2.65 | 15.09 | 17.59 |
| | Leaky ReLU | 64.20 | 61.59 | 46.81 | 28.49 | 2.61 | 14.78 | 18.32 |
| 0.4 | Linear | 68.45 | 65.75 | 49.49 | 30.28 | 2.70 | 16.26 | 19.21 |
| | Tanh | **67.75** | **66.58** | **51.03** | 29.41 | 1.17 | 15.55 | 21.62 |
| | HardTanh | 67.33 | 66.17 | 50.75 | 29.18 | 1.16 | 15.42 | 21.57 |
| | ReLU | 67.52 | 65.55 | 49.72 | 30.18 | 1.97 | 15.83 | 19.54 |
| | Leaky ReLU | 67.46 | 65.71 | 49.40 | **30.43** | 1.75 | 16.31 | 18.97 |

Table 10: Server accuracy of `Resnet44` using four different $\alpha$ and four different participation $R$, where $N = 100$. The most right columns show the accuracy drop as non-IIDness decreases.

| Participation Ratio | Activation Function | $\alpha = 10$ | $\alpha = 1$ | $\alpha = 0.1$ | $\alpha = 0.01$ | $\alpha = 10 \to 1$ | $\alpha = 1 \to 0.1$ | $\alpha = 0.1 \to 0.01$ |
|---|---|---|---|---|---|---|---|---|
| 0.1 | Linear | 49.68 | 47.28 | 39.56 | 23.26 | 2.40 | 7.72 | 16.30 |
| | Tanh | 49.13 | 46.47 | 38.19 | 23.06 | 2.66 | 8.28 | 15.13 |
| | HardTanh | **49.33** | **47.10** | **38.88** | **23.77** | 2.23 | 8.22 | 15.11 |
| | ReLU | 48.89 | 46.38 | 38.57 | 22.84 | 2.51 | 17.81 | 15.73 |
| | Leaky ReLU | 48.77 | 46.27 | 37.35 | 23.05 | 2.50 | 8.92 | 14.30 |
| 0.2 | Linear | 57.55 | 55.98 | 45.14 | 27.50 | 1.57 | 10.84 | 17.64 |
| | Tanh | 57.94 | 55.54 | **46.42** | **27.23** | 2.40 | 9.12 | 19.19 |
| | HardTanh | **58.80** | **57.54** | 45.98 | 26.46 | 1.26 | 11.56 | 19.52 |
| | ReLU | 56.89 | 54.23 | 44.42 | 26.06 | 2.66 | 9.81 | 18.36 |
| | Leaky ReLU | 57.02 | 53.76 | 43.97 | 26.37 | 3.26 | 9.79 | 17.60 |
| 0.3 | Linear | 63.29 | 61.20 | 47.78 | 30.71 | 2.09 | 13.42 | 17.07 |
| | Tanh | **64.43** | 61.27 | 50.67 | **29.53** | 3.16 | 10.60 | 21.14 |
| | HardTanh | 64.31 | **61.42** | **51.23** | 28.84 | 2.89 | 10.19 | 22.39 |
| | ReLU | 62.79 | 60.19 | 46.25 | 28.41 | 2.60 | 13.94 | 17.84 |
| | Leaky ReLU | 62.63 | 59.63 | 46.84 | 28.27 | 3.00 | 12.79 | 18.57 |
| 0.4 | Linear | 67.30 | 65.09 | 51.25 | 32.67 | 2.21 | 13.84 | 18.58 |
| | Tanh | **68.14** | 64.67 | 52.89 | **32.37** | 3.47 | 11.78 | 20.52 |
| | HardTanh | 67.91 | **65.02** | **53.95** | 31.90 | 2.89 | 11.07 | 22.05 |
| | ReLU | 66.74 | 64.20 | 50.31 | 30.51 | 2.54 | 13.89 | 19.80 |
| | Leaky ReLU | 66.41 | 64.26 | 50.40 | 31.03 | 2.15 | 13.86 | 19.37 |

## C.3 MobileNetv2 Result

Table 11 shows the result of `MobileNetv2` with four different $R$ with $N = 100$ and $\alpha = 0.1$. Tanh-like activation functions surpass recent SOTA activation functions. Especially recent SOTA activation functions accuracy is lower than Linear.

Table 11: Server accuracy of `MobileNetv2` with four different participation ratio $R$ (0.1, 0.2, 0.3, 0.4). We use $N = 100$ with $\alpha = 0.1$.

| Activation Function | CIFAR-10 | | | | CIFAR-100 | | | |
|---|---|---|---|---|---|---|---|---|
| | $R = 0.4$ | $R = 0.3$ | $R = 0.2$ | $R = 0.1$ | $R = 0.4$ | $R = 0.3$ | $R = 0.2$ | $R = 0.1$ |
| Linear | 36.74 | 36.86 | 35.86 | 35.50 | 15.02 | 14.76 | 14.23 | 13.17 |
| Tanh | 33.39 | 33.64 | **31.59** | **31.83** | **19.20** | **16.99** | **14.76** | **11.19** |
| HardTanh | 33.82 | 32.41 | 28.91 | 29.95 | 18.64 | 15.91 | 13.54 | 9.39 |
| ReLU | **36.99** | **34.21** | 26.93 | 20.12 | 16.71 | 13.97 | 8.16 | 5.17 |
| Leaky ReLU | 35.54 | 33.07 | 30.69 | 20.44 | 16.92 | 13.84 | 10.14 | 5.67 |

## C.4 FedProx Result

As mentioned in section 1, studies to improve the server model in FL add proximal terms to the local object to enhance the method's stability so that the local model rarely differs from the global model. We choose FedProx [33] as an additional FL method because it is the most basic method that focuses on improving the server model by adding a proximal term. The details of FedProx is shown in Appendix B. Table 12 shows the accuracy for different activation functions using FedProx as a learning algorithm. Similar to FedAvg, HardTanh achieves the best accuracy, followed by the other Tanh-like activation functions. Proximal terms in local training do not appear to help prevent accuracy loss.

Table 12: Server accuracy using FedProx with following setting: $N = 100$, $R = 0.2$, and $\alpha = 0.1$.

| Act. Func. | Acc. |
|---|---|
| Linear | 48.74 |
| Tanh | 45.50 |
| HardTanh | **49.16** |
| ReLU | 42.19 |
| Leaky ReLU | 42.40 |
| Swish | 35.79 |
| Mish | 39.83 |
| GeLU | 36.00 |

# D Landscape Result

We visualize the affect of different activation functions; Tanh, HardTanh, ReLU, and Leaky ReLU. The color bar beside each figure shows the loss difference. Tanh and HardTanh has extremely small number compared to ReLU and Leaky ReLU. Which emphasize that Tanh and HardTanh smooth out the landscape considerably comparison to ReLU and Leaky ReLU. Additionally, Tanh and HardTanh has its significant lowest value where perturbation X and perturbation Y is both 0. However, ReLU and Leaky ReLU does not show this phenomenon. Which we can consider that only Tanh and HardTanh succeed in reaching the global optimum.

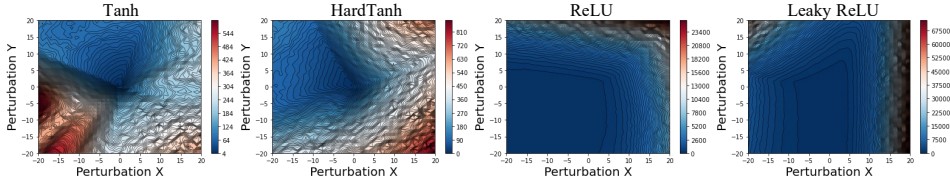

Figure 6: Landscape of `ConvNet4` with Tanh, HardTanh, ReLU and Leaky ReLU. We use $N = 100$ with $R = 0.1$ and $\alpha = 0.01$ for training. We draw landscape with 150 levels.

