# OpenReview forum: "Revisiting the Activation Function for Federated Image Classification"
_NeurIPS.cc/2022/Workshop/Federated_Learning — FL-NeurIPS 2022 Poster_

### Official Review · Reviewer_WGuX · 2022-10-07

The paper empirically evaluates the impact of the choice of activation function on federated learning in CIFAR10 and CIFAR100. In particular, it investigates whether activation functions that perform well in centralized deep learning also perform well in federated learning.

The empirical results indicate that ReLUs (and their variants), which perform very well in centralized training, perform not as well in federated training. There, tanh and HardTanH performed best. This difference is stronger for more heterogeneous local data distributions and less client participation.

The paper does not discuss the impact of hyperparameters, such as the communication period or mini-batch size on the results. From App. B.3 I understand that for FL, clients communicate every 5 epochs, and learning rate schedules are kept equal for all experiments. I assume this holds for the mini-batch size as well. I could not find the hyperparameters for the centralized setup, though. Since these parameters can have a strong impact on performance and optimal parameters can vary greatly for different setups, it would be great to optimize hyperparameters for each setup individually and - ideally - to investigate the sensitivity of the results to hyperparameter changes. For a workshop, however, the results are sufficiently interesting and noteworthy.

Strength:
- interesting to the FL community
- solid empirical analysis
- paper is well-written

Weakness:
- hyperparameters are not optimized for each setup

The paper is well-written, covers an interesting topic that fits well to the workshop, and is technically sound. Thus, I vote for acceptance.

---

### Official Review · Reviewer_v2YB · 2022-10-15
**Interesting idea, poor results.**

Summary
This paper explores the selection of activation function for federated learning. Experiments show that the optimal selection of activation function for federated learning is different to the selection for centralized learning.

Strengths
• This paper debunks the assumption that hyper-parameters that work well for centralized learning also work well for federated learning, which is important.
• Experiments cover multiple models under different hyper-parameters.

Weaknesses
• Experiments are not repeated with different random seeds. Thus, the conclusion may not be true.
• The block-wise pattern in Figure 3 looks weird. Why client 2 is similar to all other clients but client 4?
• Differences between different activation functions are not significant for widely used ResNet and MobileNet models.

---

### Decision · Program_Chairs · 2022-10-20

Accept (Poster)